# The Role of an Artificial Intelligence Method of Improving the Diagnosis of Neoplasms by Colonoscopy

**DOI:** 10.3390/diagnostics13040701

**Published:** 2023-02-13

**Authors:** Ilona Vilkoite, Ivars Tolmanis, Hosams Abu Meri, Inese Polaka, Linda Mezmale, Linda Anarkulova, Marcis Leja, Aivars Lejnieks

**Affiliations:** 1Health Centre 4, LV-1012 Riga, Latvia; 2Digestive Diseases Center GASTRO, LV-1079 Riga, Latvia; 3Department of Doctoral Studies, Riga Stradins University, LV-1007 Riga, Latvia; 4Department of Internal Diseases, Riga Stradins University, LV-1007 Riga, Latvia; 5Institute of Clinical and Preventive Medicine, University of Latvia, LV-1586 Riga, Latvia; 6Riga East University Hospital, LV-1038 Riga, Latvia; 7Faculty of Residency, Riga Stradins University, LV-1007 Riga, Latvia; 8Liepaja Regional Hospital, LV-3414 Liepaja, Latvia; 9Faculty of Medicine, University of Latvia, LV-1586 Riga, Latvia

**Keywords:** colorectal cancer, adenoma, colonoscopy, polyps, artificial intelligence

## Abstract

Background: Colorectal cancer (CRC) is the third most common cancer worldwide. Colonoscopy is the gold standard examination that reduces the morbidity and mortality of CRC. Artificial intelligence (AI) could be useful in reducing the errors of the specialist and in drawing attention to the suspicious area. Methods: A prospective single-center randomized controlled study was conducted in an outpatient endoscopy unit with the aim of evaluating the usefulness of AI-assisted colonoscopy in PDR and ADR during the day time. It is important to understand how already available CADe systems improve the detection of polyps and adenomas in order to make a decision about their routine use in practice. In the period from October 2021 to February 2022, 400 examinations (patients) were included in the study. One hundred and ninety-four patients were examined using the ENDO-AID CADe artificial intelligence device (study group), and 206 patients were examined without the artificial intelligence (control group). Results: None of the analyzed indicators (PDR and ADR during morning and afternoon colonoscopies) showed differences between the study and control groups. There was an increase in PDR during afternoon colonoscopies, as well as ADR during morning and afternoon colonoscopies. Conclusions: Based on our results, the use of AI systems in colonoscopies is recommended, especially in circumstances of an increase of examinations. Additional studies with larger groups of patients at night are needed to confirm the already available data.

## 1. Introduction

Colorectal cancer (CRC) is the third most common cancer type worldwide [1,2]. Epidemiological studies have suggested that genetic and lifestyle factors such as smoking, alcohol drinking, obesity, and low physical activity increase the risk of CRC [3].

A colonoscopy with the discovery and appropriate removal of adenomatous polyps is considered the gold standard examination that reduces the morbidity and mortality of CRC [4,5,6].

It is known that colorectal adenocarcinoma mainly develops by the adenoma–carcinoma pathway from benign adenomatous polyps of the colon. This type of cancer mostly develops from the normal, intact colon mucosa, when polyps form on its surface; these are considered as precursor lesions of CRC. Such pathological changes in the colon mucosa are formed under the influence of several factors. These include both genetic and epigenetic processes, resulting in the silencing of several tumor-suppressor genes, activation of oncogenes, and development of chromosomal instability [7].

Usually, CRC is treated surgically and, if necessary, chemotherapy is added to the treatment; however, several studies are available on drug therapy for metastatic CRC. For example, Regorafenib, which is a multi-kinase inhibitor, targets angiogenic, stromal and oncogenic receptor tyrosine kinase (RTK) [8]. Similarly, the use of lenvatinib with capecitabine and radiation in locally advanced rectal cancer is safe [9]. This type of medication is used in late stages of the disease.

CRC is considered a suitable type of cancer for the applicability of screening, as most polyps detected during colonoscopy (including adenomas and sessile serrated lesions) can be successfully removed endoscopically, thus ensuring CRC prevention [10].

Timely detection and elimination of such polyps by polypectomy protect patients from developing CRC [11].

It is known that for every 1.0% increase in the adenoma detection rate (ADR), interval CRC risks can be reduced by up to 3.0% [12].

It is known that the time of day when a colonoscopy is performed can affect the quality of the examination. This is related to the medical doctor’s fatigue, the number of examinations that he has performed during the day, as well as the medical doctor’s experience. These factors can affect the adenoma detection rate, thus also the therapy tactics, and contribute to the development of interval cancer [13]. Polyp detection rates during the day can range from 6.3% to 20% [14,15]. These data suggest that the best time for a quality colonoscopy is the morning hours.

Polyps and adenomas may not be noticed during colonoscopy due to the anatomy of the colon, the lack of experience and the fatigue of the endoscopist. The number of unnoticed polyps can reach up to 27% for various reasons [16]. The problem of unnoticed polyps and, therefore, interval CRC will also be relevant in the future, and methods are needed to help notice as many colon formations as possible [16].

As one of the options for increasing PDR and ADR, the presence of a second researcher during the screening colonoscopy could be considered, but such a strategy requires additional staff time resources, which would not be acceptable with the current shortage of medical personnel. Moreover, the research results show that such an approach is not fully effective in the context of ADR [17].

In recent years, multiple systems have been developed for automatic polyp detection assistance [18]; however, there is still not enough research on the effectiveness of such systems in real time directly in the process of clinical screening colonoscopies.

A system to be used daily during routine examinations should be one that, in terms of efficiency, is comparable to the productivity of a high-quality endoscopist (endoscopist with a high ADR). The system should be able to improve both the endoscopist’s PDR and ADR, regardless of the time of day, the endoscopist’s fatigue, and the number of examinations already performed during the day.

Artificial intelligence (AI) could be useful in reducing the possible errors of the colonoscopy specialist, to draw the medical doctor’s attention to the suspicious area of the colon mucosa [19,20].

AI-powered endoscopic systems have the potential to improve the accuracy and efficiency of diagnostic and therapeutic procedures, such as identifying and classifying polyps and other lesions. Some examples of AI-powered endoscopic systems include those that use computer vision algorithms to analyze images and videos of the gastrointestinal tract to detect and classify lesions, and those that use machine learning algorithms to assist in the interpretation of endoscopic images and videos. Additionally, AI-powered endoscopic systems can also assist in real-time during the procedure, guiding the endoscopist to the targeted area, and providing information about the tissue being examined.

The aim of this study was to compare polyp detection rate (PDR) and adenoma detection rate (ADR) in examinations performed with and without AI in the morning and afternoon hours.

## 2. Materials and Methods

This was a prospective single center randomized controlled study conducted in an outpatient endoscopy unit at Health Center 4, Riga, Latvia, with the aim of evaluating the usefulness of AI-assisted colonoscopy in PDR and ADR during daytime in the period from October 2021 to February 2022.

### 2.1. Study Population

From October 2021 to February 2022, 837 colonoscopy examinations were performed, and colonoscopies from 400 patients were included in the study. To be eligible for participation in the study, participants had to meet the following inclusion criteria: age of at least 18, and they had to have signed the informed consent form. Informed consent was obtained from all subjects involved in the study. Patients were sent for a colonoscopy examination by the family doctor; therefore the preparation schemes were different for each individual.

In turn, the exclusion criteria were: 1, a previously undergone colonoscopy examination; 2, inflammatory bowel diseases; 3, hereditary polyposis syndrome; 4, known CRC; 5, previously undergone colorectal surgery; 6, contraindications for polypectomy; 7, bad bowel preparation on a Boston-Bowel-Preparation-Scale (BBPS) of 0 to 1 in any of the three bowel segments; 8, patients with standard contraindications to colonoscopy such as acute diverticulitis and known or suspected perforation. Incomplete colonoscopies (those where endoscopists did not successfully intubate the cecum due to technical difficulties or poor bowel preparation) were excluded from the primary analysis.

### 2.2. Randomization

All eligible patients were randomized for ENDO-AID CADe AI + (study group) and ENDO-AID CADe AI—(control group). Prior to the colonoscopy procedures, patients were divided into groups. Patients were blinded to the group allocation results.

### 2.3. Study Progress

Colonoscopy examinations were performed with the Olympus Evis X1 video endoscopy system and the ENDO-AID CADe AI device.

The examinations were performed by two endoscopists, who perform an average of 2000 colonoscopy examinations per year; one endoscopist has more than 15 years of experience, and the other one has eight years.

A total of 400 patients were included in the study and randomly divided into two groups. One hundred and ninety-four patients were examined using the ENDO-AID CADe AI device (study group) and 206 patients were examined without the ENDO-AID CADe AI device (control group).

The ENDO-AID CADe AI device system works in real-time; the CADe AI was used during the evacuation of the instrument from the caecum, marking the found polyp on the main monitor with a green outline (Figure 1, Figure 2, Figure 3, Figure 4 and Figure 5), drawing the endoscopist’s attention to changes of the mucosa. Cases in which the AID CADe AI device showed a polyp but the endoscopist did not detect it were defined as a false positive system response.

Due to patient data security concerns, endoscopic images were obtained through Olympus.

It was assumed that morning examinations are carried out between 8.00 and 13.00, and afternoon examinations are between 13.00 and 18.30.

### 2.4. Colonoscopy Procedure

Colonoscopy examinations were performed with the Olympus EVIS V1 video endoscopy system and the ENDO-AID CADe artificial intelligence device. All colonoscopy examinations were performed under anesthesiologist supervision under short-term intravenous isolated propofol sedation. The dose of medicine was determined by the anesthesiologist.

Bowel cleanliness was assessed by an endoscopist using the BPPS scale. Six subjects were excluded from the study due to poor bowel preparation (0–1 in any of the three bowel sections).

The time of evacuation of the instrument from the cecum for each performed colonoscopy was not less than 7 min and was monitored by the endoscopist’s assistant.

Any detected polyp was described in the colonoscopy report according to the Paris [21] and NICE [22] classifications, and the location of the polyp in the colon and its size were specified.

The biopsied or ablated polyp was sent for morphological examination. If a polyp was found, at least two pieces from the lesion were taken before polypectomy; in the event that the removed polyp disappeared in the colon fluid, the tissue was available for morphological evaluation.

The examination begins by inspecting the anal canal and the rectum, performing an examination in retroversion to evaluate the mucosa above the “Z” line and evaluate the condition of internal hemorrhoidal nodes. When the mucosa of the anal canal is viewed in retroversion, the endoscope is advanced proximally into the sigmoid, descending, transverse, and ascending the colon until the dome of the cecum is reached. By moving the endoscope proximally into the colon, the residual contents are evacuated from the colon. A mandatory requirement for a high-quality colonoscopy examination is photo-documentation of the opening of the appendix. If technically possible, a retroversion examination was also performed in the cecum.

The right side of the colon was examined twice, returning distally from the cecum to the transverse colon, inspecting the mucosa in NBI mode, and again proximal to the cecum.

In order to detect flat polyps of the right colon, after evacuating the instrument from the caecum for the second time, staining of the mucosa with methylene blue solution was performed.

Further inspection of the mucosa of the colon was performed by gradually evacuating the endoscope, also carefully inspecting the mucosa behind the folds.

### 2.5. Morphological Diagnosis of Lesions Found during Colonoscopy

All removed polyps and specimens from biopsied lesions were transmitted to the Academic Histology laboratory (Riga, Latvia) for morphological diagnostics. All samples were analyzed by expert pathologists. All removed and biopsied lesions were analyzed and characterized according to the World Health Organization criteria, depending on morphological characteristics [23]. All lesions were described as serrated polyps and lesions, low-grade dysplasia (LGD), high-grade dysplasia (HGD), superficial submucosal invasive carcinoma (SM-s; <1000 μm of submucosal invasion) and deep submucosal invasive carcinoma (SM-d; ≥1000 μm of submucosal invasion). No traditional serrated adenoma (TAS), sessile serrated lesion with dysplasia (SSL-D) or unclassified serrated adenoma were found morphologically.

### 2.6. Statistical Analysis

For baseline sociodemographic and clinical characteristics and colonoscopy quality parameters, comparison between the two groups was performed using the Mann-Whitney-U-test for continuous variables and the Chi-square test for categorical variables. A *p*-value of less than 0.05 was considered to signify statistical significance. Statistical analysis was performed using IBM SPSS (version 20) [24].

## 3. Results

During the study period, two gastroenterologists performed and analyzed 400 colonoscopies. Participant age ranged from 18–84 years. Our results showed that there are no statistical differences regarding age and gender between the two groups (overall age of study participants: 50.7 ± 14.9 years, 193 males, 207 females were included; AI−: 51.2 ± 14.5 years, 102 males, 104 females; AI+: 50.1 ± 15.4 years, 91 males, 103 females). First doctor (DR1) performed 98 non-AI-assisted colonoscopies (42 in the morning and 56 in the afternoon) and 91 AI-assisted colonoscopies (43 in the morning and 48 in the afternoon); while the second doctor (DR2) performed 108 non-AI-assisted colonoscopies (62 in the morning and 46 in the afternoon) and 103 AI-assisted colonoscopies (57 in the morning and 46 in the afternoon). In total, the study group had 194 colonoscopies, while the control group had 206 colonoscopies (Table 1).

### 3.1. Polyp Detection Rates (PDR)

In the morning PDR without AI for DR 1 was 33.3%, but with AI it was 32.6% (*p* = 0.939), while in the afternoon PDR without AI was 28.50% and with AI 37.50% (*p* = 0.333), which indicates that adding AI during afternoon colonoscopies for (DR1) improves PDR, but not statistically significantly (Table 2 and the 4th Table in Section 3).

Analyzing DR2 results, the morning PDR without AI was 59.7%, and with AI it was 54.4% (*p* = 0.56). Similarly to DR 1, PDR in morning colonoscopies for DR 2 without AI was also higher, but not statistically significantly; however, afternoon PDR without AI was 39.10%, while with AI assistance it increased to 50% (*p* = 0.256), which indicates that PDR increases significantly when using AI, but not statistically significantly (Table 3 and Table 4).

### 3.2. Adenoma Detection Rate (ADR)

In the morning, ADR without AI for DR 1 was 16.3%, but with AI it was 27.9% (*p* = 0.214), which confirms the beneficial effects of AI in increasing ADR for this doctor also during morning colonoscopies, but not statistically significantly (Table 2 and Table 4).

In the afternoon, ADR without AI for DR1 was 14.3% and with AI it was 22.90% (*p* = 0.256), indicating that adding AI during afternoon colonoscopies also improves ADR for DR 1, but not statistically significantly (Table 2 and Table 4).

For DR2, ADR in the morning without AI assistance was 29%, and with AI assistance it increased to 35.10% (*p* = 0.479), while in the afternoon ADR without AI was 21.70%. Using the AI function, it increased to 34.8% (*p* = 0.165), which indicates that the use of AI during afternoon colonoscopies significantly improves ADR for DR 2, but not statistically significantly (Table 3 and Table 4).

## 4. Discussion

Although in recent decades CRC mortality and incidence rates have significantly decreased (51% and 32%, respectively) [25], CRC is still an urgent problem of the health care system worldwide with relatively high incidence and mortality rates. The reduction of these indicators is mainly related to the introduction of CRC screening in several countries, as well as the improvement of the quality of colonoscopy examinations and the appropriate qualitative removal of adenomas [25].

Considering that the realization of the adenoma-carcinoma pathway requires a relatively long time, as well as the fact that early-stage CRC remains mostly asymptomatic, it is possible to treat it preventively with the help of polypectomies and to apply highly effective treatment in the early stages of an already known oncological disease. Current knowledge about the pathogenesis of CRC and the natural course of the disease suggests that CRC and precancerous changes in the bowel are suitable conditions for screening [10].

To ensure the highest possible quality of the colonoscopy examination, the preparation of the colon before the examination with a split regimen is still relevant. Sufficient time, not less than 6 min, but preferably at least 8 min, should also be devoted to the evacuation of the instrument from the caecum. If retroflection on the right side of the colon is not possible, then this area should be examined twice—going from the caecum to the transversum, then again to the caecum, and then evacuating the instrument completely, devoting time to other segments of the colon as well. This approach improves ADR scores [26].

In recent years, the method of colonoscopy has progressed significantly. Endoscopists can improve colonoscopy quality and ADR by using specific tools during routine examinations—for example, various endo cuffs for expanding the field of view of the mucosa and straightening folds of the colon, digital chromoendoscopy options for more successful detection of flat polyps and determination of borders, so that the result of a polypectomy achieves maximum value. Full spectrum colonoscopy options are also available, as are devices that assist retrograde examination during a colonoscopy [27].

All the previously described methods allow the colonoscopy examination to become as effective as possible, thus contributing to the 5-year survival rate in CRC [25].

Even using all the methods described above to improve PDR and ADR, some polyps remain undetected and unremoved. More often, undetected polyps hide behind the folds of the bowel and are not in the field of view of the endoscopist. However, other data reveal that in some cases, especially if the polyp is flat or serrated, it can go unnoticed even when it lies in the field of view of the endoscopist [28].

Such errors can be observed when the colonoscopy is performed by an endoscopist without sufficient experience and qualifications, as well as when the endoscopist’s attention is diverted from the screen for unclear reasons [29].

Similarly, errors during colonoscopy should be discussed in the context of the researcher’s fatigue, burnout, and the time of the working day when the examination is performed.

To contribute to the further reduction of CRC incidence and mortality, the implementation of an automatic polyp detection system in daily practice should be considered. However, it should have very high sensitivity and specificity in real time to maintain a sufficiently high ADR and avoid false positives [30].

The results of research conducted in 2019 showed a significant increase in ADR, PDR, and mean number of polyps and adenomas per colonoscopy in the CADe group compared to the control group. However, it should be recognized that this increase was mainly due to an increase in the number of adenomas smaller than 5 mm. It is known that these diminutive lesions are often the most difficult to find endoscopically. It is also known that small adenomas are associated with a lower risk of developing a malignant tumor. Therefore, one would think that such findings could lead to additional unnecessary polypectomies and an increased workload. However, finding and eliminating any adenoma during colonoscopy could reduce the risks of developing CRC.

It should be noted that polyps that are not in the doctor’s field of view during the colonoscopy are still a major drawback of the current CADe system. Additional technologies that would able to deal with the detection of polyps behind the folds of the colon, in hard-to-see areas of the mucosa, are needed [31].

### The Role of Artificial Intelligence in Colonoscopy Examinations

The functioning of the brain and the subsequent provision of all body functions is still not fully understood. Currently, people mainly use their visual recognition functions in the context of endoscopic image perception, but this type of image perception has its limitations [32].

So far, data have been obtained that the human brain and its vision and image recognition functions are not fully capable of processing when compared to computer image perception capabilities [33]. In addition, an independent imaging operator could be a good aide for doctors, especially those with little experience and seniority, as well as in situations where an experienced doctor with a lot of work experience feels tired or overworked. Recently, artificial intelligence—a tool that is in a constant self-learning process (machine learning)—is increasingly being considered as such an independent assistant; therefore, it has the potential to solve more difficult problems, including in medicine and gastrointestinal endoscopy. Machine learning as a method creates mathematical algorithms in automatic mode that are based on the data offered by the specialist or on training data; therefore, the algorithm can assist a person, and in certain circumstances even make decisions without human intervention [34].

Artificial intelligence (AI) is currently an assistive device in a constant learning process, which provides solutions that are tied to human intelligence by storing and processing the acquired data. Basically, types of artificial intelligence are designed to analyze visual images; however, there are also developments that recognize and process data related to audio, voice and various languages [35].

Currently available methods mainly use Deep Neural Networks (DNN) and Convolutional Neural Networks (CNN). These networks have the ability to independently derive data from, for example, available healthcare data and thereby augment their work capabilities and functions [36].

It is in the context of colonoscopies that two important roles of artificial intelligence in maintaining the quality of the examination have crystallized. The first is the detection of polyps during the examination (CADe), when the device informs the endoscopist about the presence of a mucosal lesion with both an audible and a visual signal, encircling the suspicious mucosal area with a green border. The use of this function would have the potential to contribute to an increase in ADRs for each endoscopist, and it is likely that the CADe function would make the greatest contribution during afternoon examinations.

The second feature of the artificial intelligence device that could be used during routine examinations is polyp characterization (CADx). The use of this function would allow the endoscopist to improve the accuracy of the optical diagnosis and to navigate the morphological structures of a particular polyp already during the examination, thus predicting whether the detected polyp is adenomatous or not. This would allow the examiner to avoid performing unnecessary non-adenomatous polypectomies, thereby avoiding the development of additional complications, as well as saving significant amounts of the health care budget. It is known that avoiding the removal of unnecessary non-adenomatous masses could save up to $33 million annually in the US [30].

The results of our study indicate that in afternoon colonoscopies, both PDR and ADR are lower for both endoscopists compared to morning colonoscopies. Most likely, this finding could be related to the doctor’s fatigue and reduced ability to concentrate in the second half of the working day.

It is known that having long working hours and performing many examinations during the day can reduce the cognitive abilities of the doctor and contribute to the frequency of the development of medical errors [37].

The results of our study could also be applied to endoscopic examinations; that is, the quality of performed colonoscopies could also be related to the doctor’s fatigue during the day.

Scientists have tried to measure the degree of fatigue in the context of colonoscopy examinations by using the time spent at work [38] but also by evaluating the work intensity directly in terms of the number of examinations performed [39]. The most common indicators used to evaluate the quality of a colonoscopy are ADR and the cecal intubation rate (CIR) [40].

Overwork of doctors has been a hot topic in recent years [37].

It is known that, for example, the demand for colonoscopy examinations has grown significantly in China, and thus the fatigue of endoscopists due to work overload is also increasing [41].

Several studies have not shown a negative effect of a higher number of examinations on ADRs [39,42]. However, after the sixth procedure, a drop in CIR is observed, which could also be related to endoscopist fatigue [42]. In the same way, the results of the study indicate that the ADR during afternoon colonoscopies is lower compared to examinations performed in the morning [4].

Available research [43] also indicates that each hour worked has a negative impact on the ADR result. Solutions are being sought to maintain the consistently high quality of colonoscopy examinations even in situations when the doctor feels tired or overworked, and more and more attention is being paid to the possible use of automatic polyp detection systems daily.

When using AI-assisted colonoscopies in the morning examinations for both doctors, the PDR was lower than in colonoscopies without AI. However, this does not apply to ADR morning colonoscopies. These results could indicate higher concentration abilities of doctors in the morning.

Considering that the development of CRC is mainly associated directly with adenomatous polyps, ADR is clearly a more important indicator, and it increases in both morning and afternoon colonoscopy examinations for both doctors, but not statistically significantly.

Future studies could consider analyzing the degree of physician fatigue using other objective, independent tools, which could provide more reliable information on the association of endoscopist fatigue with the reduction of ADRs during the day.

Considering that there are data showing that an additional expert participating in the endoscopy procedure improves both PDR and ADR [44], the daily use of AI-assisted colonoscopy in endoscopy rooms could reduce the workload of staff, maintaining adequate quality of colonoscopies with sufficient PDR and ADR for days at a time.

Data from other studies on ADR exposure during weekday afternoons are also available [45].

However, there are also reports that colonoscopies performed in the afternoon were of better quality, specifically in the context of ADR [46]. It is thought that this is more related to the preferences of the specific doctor’s daily rhythm, as well as to whether the specific doctor worked a full day or only an afternoon shift; such data are reflected in the meta-analysis of Wu et al. performed in 2018 [47].

The limitations of our work include the fact that the study was conducted in one center, and the colonoscopies were performed by only two endoscopists. It should also be considered that if the number of patients included in the study were larger, statistically reliable differences in the results between the study and control groups could be expected.

In addition to the already existing information, data on colonoscope evacuation time, which is also known to be a significant factor in the raising of ADR, would probably be useful.

Based on the results of our study, the use of AI systems in routine screening colonoscopies is considered justified and recommended, especially in circumstances where there is an increase in the number of diagnostic examinations. In the future, however, attention should be paid to the problem of the lack of endoscopists, the workload of doctors, and whether colonoscopies are performed within endoscopy departments’ quality requirements. Additional studies with larger groups of patients during the working day are needed to confirm the already available data.

## 5. Conclusions

The results of this study indicate that, although there is no statistically significant difference between groups, the real-time CADe-assisted system increases the PDR during afternoon colonoscopies for both physicians, as well as the ADR during both morning and afternoon colonoscopies for both physicians. Thus, CADe-assisted systems should be considered for use during colonoscopies in routine work.

## Figures and Tables

**Figure 1 diagnostics-13-00701-f001:**
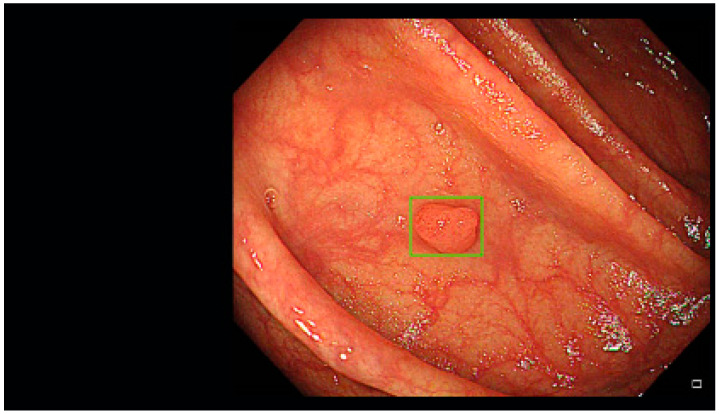
Polyp detected by CADe AI.

**Figure 2 diagnostics-13-00701-f002:**
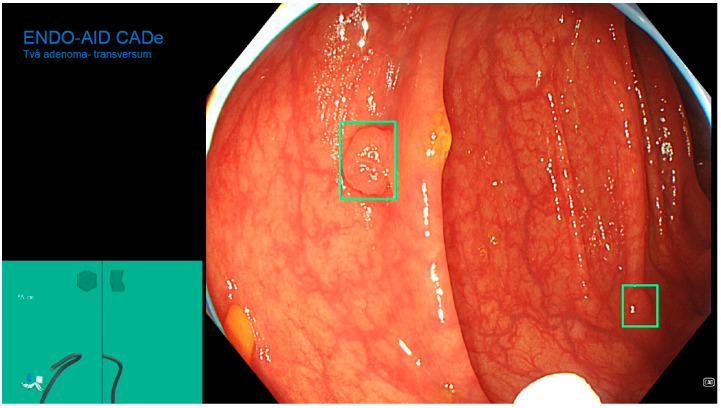
Two polyps detected by CADe AI.

**Figure 3 diagnostics-13-00701-f003:**
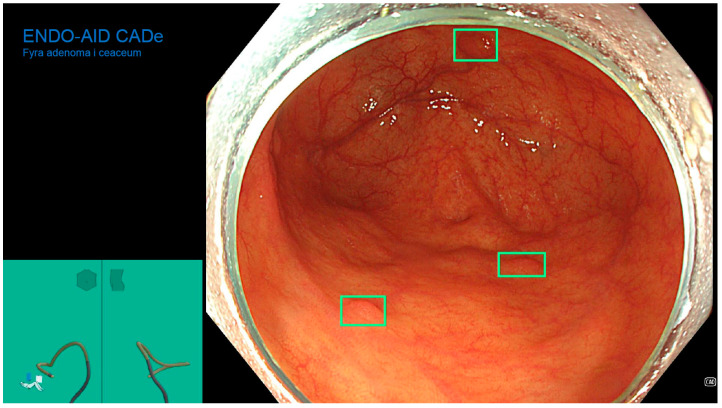
Three polyps detected by CADe AI.

**Figure 4 diagnostics-13-00701-f004:**
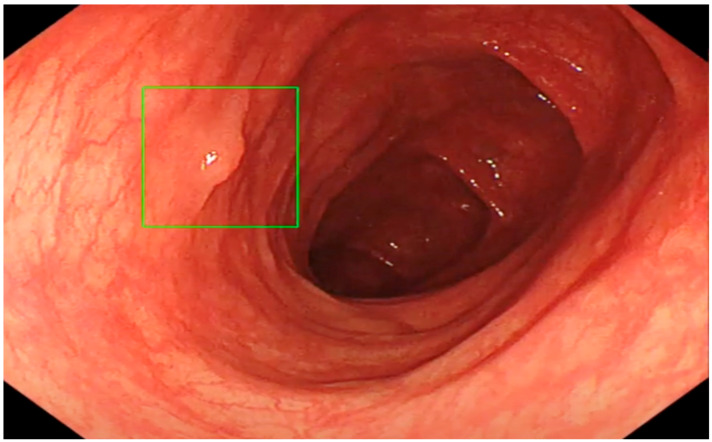
Diminutive polyp detected by CADe AI.

**Figure 5 diagnostics-13-00701-f005:**
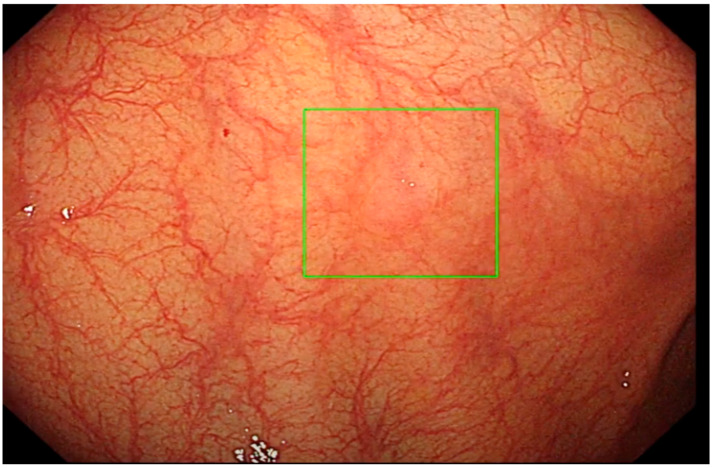
Flat polyp detected by CADe AI.

**Table 1 diagnostics-13-00701-t001:** Overall patient characteristics.

	Without AI	With AI	Overall
Females	104	103	207
Males	102	91	193
DR1			
Patients	98	91	189
Morning	42	43	85
Afternoon	56	48	104
DR2			
Patients	108	103	211
Morning	62	57	119
Afternoon	46	46	92
Overall			
Patients	206	194	400

**Table 2 diagnostics-13-00701-t002:** The results of DR 1 examinations.

	Morning Time		*p*-Value	Afternoon		*p*-Value
	Without AI	With AI		Without AI	With AI	
Colonoscopies	42	43		56	48	
Polyps found	14	14	0.939	16	18	0.333
PDR	33.3%	32.6%		28.5%	37.5%	
Adenomas found	7	12	0.214	8	11	0.256
ADR	16.3%	27.9%		14.3%	22.9%	

DR 1—first doctor, AI—artificial intelligence, PDR—polyp detection rate, ADR—adenoma detection rate.

**Table 3 diagnostics-13-00701-t003:** The results of DR 2 examinations.

	Morning Time		*p*-Value	Afternoon		*p*-Value
	Without AI	With AI		Without AI	With AI	
Colonoscopies	62	57		46	46	
Polyps found	37	31	0.56	18	23	0.294
PDR	59.70%	54.40%		39.10%	50%	
Adenomas found	18	20	0.479	10	16	0.165
ADR	29%	35.10%		21.70%	34.80%	

DR 2—second doctor, AI—artificial intelligence, PDR—polyp detection rate, ADR—adenoma detection rate.

**Table 4 diagnostics-13-00701-t004:** The results of the examinations performed by both doctors.

	Morning Time		*p*-Value	Afternoon		*p*-Value
	Without AI	With AI		Without AI	With AI	
Colonoscopies	104	100		102	94	
Polyps found	51	45	0.563	34	41	0.139
PDR	49.04%	45.00%		33.33%	43.62%	
Adenomas found	25	32	0.205	18	27	0.065
ADR	24.04%	32.00%		17.65%	28.72%	

AI—artificial intelligence, PDR—polyp detection rate, ADR—adenoma, detection rate.

## Data Availability

Not applicable.

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
