# Peer review of "The Role of an Artificial Intelligence Method of Improving the Diagnosis of Neoplasms by Colonoscopy"

_diagnostics, 2023, doi:10.3390/diagnostics13040701_

Round 1
Reviewer 1 Report
1. In abstract, the author should classify the aim and the significance of this manuscript.
2. Keywords should include Artificial intelligence
3. In introduction, the introduction on AI was insufficient.
4. In table1 and table 2, the size of dataset seems small. How to ensure that the model does not overfit?
5. In 3.2, “which indicates that the use of AI during afternoon colonoscopies significantly improves ADR for DR2, but not statistically reliably (Table 3, Table 4).” If it’s possible to explain such phenomenon?
6. Only 47 literatures were cited. It seems not enough.
7. The English expressions of this manuscript should be improved
Author Response
- In abstract, the author should classify the aim and the significance of this manuscript.
The aim of evaluating the usefulness of AI-assisted colonoscopy in PDR and ADR during the day time. It is important to understand how much already available CADe systems improve the detection of polyps and adenomas in order to make a decision about their routine use in practice.
- Keywords should include Artificial intelligence
will be corrected in the text
- In introduction, the introduction on AI was insufficient.
AI-powered endoscopic systems have the potential to improve the accuracy and efficiency of diagnostic and therapeutic procedures, such as identifying and classifying polyps and other lesions. Some examples of AI-powered endoscopic systems include those that use computer vision algorithms to analyze images and videos of the gastrointestinal tract to detect and classify lesions, and those that use machine learning algorithms to assist in the interpretation of endoscopic images and videos. Additionally, AI-powered endoscopic systems can also assist in real-time during the procedure, guiding the endoscopist to the targeted area, and providing information about the tissue being examined.
- In table1 and table 2, the size of dataset seems small. How to ensure that the model does not overfit?
No machine models were trained in the study, the doctors used commercially available software with AI.
- In 3.2, “which indicates that the use of AI during afternoon colonoscopies significantly improves ADR for DR2, but not statistically reliably (Table 3, Table 4).” If it’s possible to explain such phenomenon?
Perhaps there is a larger effect of afternoon fatigue on DR2, and if the number of patients had been larger, a statistically reliable difference might have been achieved.
- Only 47 literatures were cited. It seems not enough.
The submission rules asked for at least 30 references
- The English expressions of this manuscript should be improved
will be corrected in the text
Reviewer 2 Report
Interesting article. AI is future which cannot be denied and it will be in all the fields of medicine.
Author Response
Thank you for rating our work!
Reviewer 3 Report
The article has been written well with enough information. The subject of this article is interesting for a broad range of authors.
Among all kinds of cancers, colorectal cancer (CRC) is the third most common cancer worldwide. In the introduction, there should be more references related to how to treat this cancer with drugs. There are some commercial drugs such as pazopanib, regorafenib and lenvatinib (especially regorafenib). https://link.springer.com/article/10.1007/s11030-022-10406-8 and https://www.tandfonline.com/doi/abs/10.1517/13543784.2016.1161754.
Author Response
Several studies are available on drug therapy for metastatic CRC. For example, Regorafenib, which is a multi kinase inhibitor targets angiogenic, stromal and oncogenic receptor tyrosine kinase (RTK).
Xu D, Liu Y, Tang W, Xu L, Liu T, Jiang Y, Zhou S, Qin X, Li J, Zhao J, Ye L, Chang W, Xu J. Regorafenib in Refractory Metastatic Colorectal Cancer: A Multi-Center Retrospective Study. Front Oncol. 2022 Mar 30;12:838870. doi: 10.3389/fonc.2022.838870. PMID: 35433423; PMCID: PMC9007238.
Similarly, the use of lenvatinib with capecitabine and radiation in locally advanced rectal cancer is safe.
Mehta R, Frakes J, Kim J, Nixon A, Liu Y, Howard L, Martinez Jimenez ME, Carballido E, Imanirad I, Sanchez J, Dessureault S, Xie H, Felder S, Sahin I, Hoffe S, Malafa M, Kim R. Phase I Study of Lenvatinib and Capecitabine with External Radiation Therapy in Locally Advanced Rectal Adenocarcinoma. Oncologist. 2022 Aug 5;27(8):621-e617. doi: 10.1093/oncolo/oyac003. PMID: 35325225; PMCID: PMC9355805.
This type of medication is used in the late stages of the disease.